# Biomedical open source software: Crucial packages and hidden heroes

Eva Maxfield Brown[1◉], Stephan Druskat[2◉], Laurent Hébert-Dufresne[3◉], James Howison[4◉*], Daniel Mietchen[5,6◉], Andrew Nesbitt[7◉], João Felipe Pimentel[8◉], Boris Veytsman[9,10◉]

1 Information School, University of Washington, Seattle, Washington, United States of America, 2 Institute of Software Technology, German Aerospace Center (DLR), Berlin, Germany, 3 Department of Computer Science, University of Vermont, Burlington, Vermont, United States of America, 4 School of Information, The University of Texas at Austin, Austin, Texas, United States of America, 5 Leibniz Institute for Information Infrastructure, FIZ Karlsruhe, Berlin, Germany, 6 Institute for Globally Distributed Open Research and Education (IGDORE), Jena, Germany, 7 Ecosyste.ms, London, United Kingdom, 8 Instituto de Computação, Universidade Federal Fluminense, Niterói, Rio de Janeiro, Brazil, 9 Chan Zuckerberg Initiative, Redwood City, California, United States of America, 10 School of Systems Biology, George Mason University, Fairfax, Virginia, United States of America

◉ These authors contributed equally to this work.
* jhowison@ischool.utexas.edu

## Abstract

Despite the importance of scientific software for research, it is often not formally recognized nor rewarded. This is especially true for foundational libraries, which are hidden below packages visible to the users (and thus doubly hidden, since even the packages directly used in research are frequently not visible in the paper). Research stakeholders like funders, infrastructure providers, and other organizations need to understand the complex network of computer programs that contemporary research relies upon. In this work, we use the CZ Software Mentions Dataset to map the upstream dependencies of software used in biomedical papers and find the packages critical to scientific software ecosystems. We propose centrality metrics for the network of software dependencies, analyze three ecosystems (PyPI, CRAN, Bioconductor), and determine the packages with the highest centrality.

## Author summary

Scientists today use software as an important tool for their research. The progress of science depends on the people who write software being properly recognized, rewarded, and incentivized. This means that those who fund research or promote scientists need to know the impact of software and identify the most important programs. The universe of scientific software is complex: some people write programs used by researchers, while some people write programs used by other programs. The latter are often even less visible than the former, but they are important for the progress of science.

**Data availability statement:** Name: SoftwareImpactHackathon2023_Tracing_ dependencies https://github.com/borisveytsman/SoftwareImpactHackathon2023_Tracing_dependencies doi:10.5281/zenodo.10031223.

**Funding:** The authors are grateful to the Chan Zuckerberg Initiative for making this research possible and for sponsoring the Mapping the Impact of Research Software in Science hackathon. L.H.-D. acknowledges support from the National Institute of General Medical Sciences under the 2P20GM125498 Centers of Biomedical Research Excellence Award and from Google Open Source through the Open-Source Complex Ecosystems And Networks (OCEAN) project. D.M. acknowledges support from the German Research Council (DFG) through the MaRDI project (460135501) as well as through the BASE4NFDI/KGI4NFDI project (521453681). J.F.P. acknowledges support from CNPq and FAPERJ (E-26/210.478/2024). J.H. acknowledges that this material is based upon work supported by the US National Science Foundation under Grant Nos. SMA-1064209 (SciSIP), OCI-0943168 (VOSS) and ACI-1453548 (CAREER) as well as support from the Alfred P. Sloan Foundation, Grant/Award Number: 2016-7209 for the creation of the SoftCite dataset. The Chan Zuckerberg Initiative and the acknowledged funders had no role in study design, data collection and analysis, decision to publish, or preparation of the manuscript.

**Competing interests:** The authors have declared that no competing interests exist.

In this work, we establish metrics to measure the impact of both types of software. We introduce the network of dependencies (program A used by researchers depends on programs B, C, and D) and use it to calculate the impact of a large number of packages used in the biomedical literature. In doing so, we identify important software packages, regardless of their direct visibility to end users. We conclude with a discussion of the limitations of our methodology and approaches to improve network validity.

## 1 Introduction

Since the second half of the last century, a computer has become as ubiquitous a tool of a scientific lab as balance scales and a Bunsen burner were in previous ages. As a consequence, computer software is now crucial to research, bringing new methods and new scale, while offering new potential for reproducibility and extension. This is true not only for the natural sciences and mathematics but for scholarship more broadly (e.g., sociology and the humanities), making the software revolution both wide and deep. Yet we have very limited insight into the software actually used in research. This lack of infrastructural understanding means we are limited in our ability to reward developers and maintainers, encourage collaboration and coordination, and direct science funding in a well-informed manner.

Scientific software is often invisible in publications, because citation practices in science have not changed at the same pace as software has become crucial. For example, software is infrequently and inconsistently formally cited [1–6]. There have been recent efforts to extract informal citations from the full text of articles [7–10] and to evaluate the "importance" of software packages by looking at papers that cite them [11]. Unfortunately, publications sometimes do not mention all the software used in the course of research.

Besides this, there is another kind of invisibility of scientific software. The programs visible to the end user may rely on many other software packages (known in the software world as dependencies). While the end users may mention a package at the top of the dependency stack, they are likely not even aware of the full set of packages that are further below. These may be packages that the user-facing program depends on directly (*direct dependencies*), or indirectly, where the direct dependencies in turn may depend on other packages. These latter packages thus become indirect, *transitive dependencies* of the user-facing program. This complexity in the network of dependencies has a number of implications, including those for security [12,13] and computational reproducibility [14]. In particular, much of the work undertaken to develop, maintain, test, and distribute the underlying software is not directly visible in the publication record itself, and is not included in derivatives such as citation networks and knowledge graphs.

The situation resembles the famous XKCD cartoon where "all modern infrastructure" critically depends on "a project some random person in Nebraska has been thanklessly maintaining since 2003" [15]. The word "thanklessly" is important in this

context: being unknown, these critical pieces of software get much less recognition and credit than they deserve—and than the science needs. The absence of recognition may lead to dire consequences, for example, if, due to the lack of funded maintenance, the underlying libraries become vectors for malware attacks.

There have been proposals to assign credit to these packages using the dependency structure to calculate what has been termed *transitive credit* [16,17] but so far, the problem has not yet been solved. To implement the proposed measure, software projects must start to publicize the packages they rely on in a way that enables recognition and citation, beyond the technical dependency already recorded in manifest files such as `pyproject.toml`, `DESCRIPTION`, `cargo.toml`, `pom.xml`, etc. One way for projects to do this is the inclusion of citation information for their own software outputs as well as for their direct dependencies, e.g., a citation file in the Citation File Format (CFF) [18]. Were such citation information available for the complete dependency stack of a program, transitive credit could be implemented by building weighted software citation networks [19].

At present, though, the situation is quite different: sometimes even the maintainers of the lower-stack software are not aware of the upper-stack programs that depend on their work. They need this knowledge when making breaking changes to their packages, which might negatively influence the software that depends on them [20].

By classifying software packages as visible to the end users or primarily important for other packages, we follow the ideas described by Donald E. Stokes in his famous book, Pasteur's Quadrant [21]. There, Stokes distinguishes between applied research with the results visible to the general public, and pure research, which is less visible, but important for the applied research. He discussed a two-dimensional diagram of scientific works, with the importance of the works for the applications on the *x* axis and the quest for fundamental understanding on the *y* axis. Thus, the upper left quadrant is occupied by the purely fundamental research personified by Niels Bohr. The lower right quadrant is occupied by purely applied research personified by Thomas Edison. The upper right quadrant is occupied by the works that have both fundamental and practical value. Stokes chose Louis Pasteur as an example of such works. While this picture is rather simplified (Bohr's works have significant practical value, and Edison's experiments stimulated fundamental research), it gives important insight into scientific endeavor.

Following these ideas, we can put the software packages into a two-dimensional plane with the axes corresponding to the frequency of software mentions and a measure of network centrality (Fig 1). The majority of the packages will likely occupy the lower left corner of the plot, having a small number of mentions from the authors of scientific papers and limited centrality from usage in other software packages. Popular packages are used by many authors. We are interested in the "Nebraska" packages, not very visible, but of critical importance. We use the term "Nebraska" packages throughout this article in reference to the original XKCD cartoon which noted that much of highly critical software is often less visible and less well known [15]. Of course, one can think about highly visible and critical "Pasteur" packages, which are used both directly and as a foundation for other libraries.

We had a fortunate opportunity to explore these ideas due to the generosity of the Chan Zuckerberg Initiative (CZI). On 24–27 October 2023, CZI hosted a hackathon on *Mapping the Impact of Research Software in Science* (see https://github.com/chanzuckerberg/software-impact-hackathon-2023). One of the projects at this hackathon was *Tracing the dependencies of open source software mentioned in the biomedical literature* [22]. In this project, we explored the dependencies of the open source software packages mentioned in the CZ Software Mentions dataset [9]. The dataset was created using a machine learning system trained on the SoftCite gold standard dataset of manually annotated software mentions [8] and consists of extractions from 2.4 million biomedical papers. The methodology and its limitations are discussed in [10]. In particular, the work on the disambiguation of packages can be improved. Later, we discuss how these limitations influence our results.

We decided to limit our study to open-source packages: first, because the dependencies of closed-source packages are not public, and second, because we believe in the importance of open source for open and reproducible science. Here, we explore approaches to making the software package infrastructure underlying science more visible. We examine

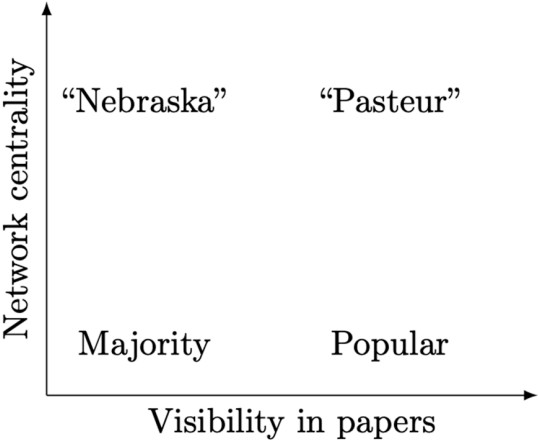

**Fig 1. Classification of software packages inspired by Stokes' classification system in [21].** "Nebraska" packages are software projects which have few mentions in research articles, but are highly central in a dependency network. "Pasteur" packages are both highly visible with lots of mentions and are highly central in a dependency network.

our findings to develop questions to better understand the idea of criticality and opportunities for improving science through adjustments to the research software ecosystems.

## 2 Materials and methods

### 2.1 Network construction

We combined three datasets: The CZ Software Mentions dataset [9], which was constructed using a machine learning model trained on the SoftCite Gold Standard annotations [8], the Ecosyste.ms software dependency dataset [23], which is built by gathering and normalizing the dependency information from multiple software ecosystems, including one for the Python programming language (PyPI) and two for the R programming language (CRAN and Bioconductor), and finally we used OpenAlex [24] for information on how frequently papers had been cited. The software authorship data were scraped from GitHub metadata.

The CZ Software Mentions dataset [9,10] includes mentions from 2.4 million biomedical papers and identifies which mentions were traced to which ecosystem. In this way, package names from CRAN, Bioconductor, and PyPI are parsed by the software mentions extraction model, collected, and finally disambiguated. We collected information for each package, including its dependencies, using the latest release of the package at the time of data collection (October 2023). Dependencies were then recursively retrieved using the most recent release and following dependencies until the full list of transitive dependencies was obtained. Limitations of our dependency resolution approach are discussed later.

After this data processing, we produced a two-mode network, with nodes for papers and nodes for software packages. Edges from papers to software packages were added when a package was mentioned in the full text of the article. Edges between packages were added when metadata descriptions indicated a required dependency. The full code for the processing is available at [22]. The network is available at [25] in GEXF format [26] and has four classes of nodes:

**paper:** papers from the CZ Software Mentions dataset. They are identified by their Digital Object Identifier (DOI). To estimate the impact of the papers, we separately downloaded their citation numbers from OpenAlex [24] as of November 2023 (a copy is available in the data subdirectory of [22]).

**pypi, cran, bioconductor:** Software from the corresponding ecosystems. We used CZI ID [9,10] as the identifier. We did not attempt to identify the same software across the ecosystems (see the discussion below).

   

The edges are directed and weighted. An edge from one software node *A* to another software node *B* means that software *A* depends on software *B* as determined by the corresponding metadata. An edge from a paper node to a software node means that the given software is mentioned in the paper, and the weight corresponds to the number of citations the paper received. In what follows, we will consider both the weighted and unweighted versions of this network, as utilization of the weights has both positive and negative aspects to consider during analysis and interpretation.

### 2.2 Network analysis

To analyze the network, we relied on directed centrality analysis [27]. There are several possible options for centrality measures. Our choice was determined by the following considerations. First, we wanted a centrality metric that can account for papers even if they have no incoming edges in our network (in other words, papers do not receive centrality but should contribute to the centrality of software packages). This criterion excludes eigenvector centrality [28]. Second, we wanted a centrality metric that can account for the weight of edges from papers to software packages (i.e., some papers contribute more centrality because they are more highly cited.). This criterion excludes the PageRank algorithm [29], which normalizes weights of out-degree. We found that Katz centrality [30] satisfies all our criteria. Katz centrality can be interpreted as the importance of the node for the diffusion processes on the network. This corresponds to the intuition that innovation is a diffusion-like process [31]. Katz centrality gives us the attenuation factor $\beta$ as a free parameter to control the importance of papers (which we set equal to 1), such that paper nodes contribute a factor proportional to $\beta$ times their citation count to the software packages they mention (we follow the formulation of Katz centrality in [32]). In turn, software packages contribute a factor proportional to their own centrality to their dependencies. A package can therefore be central by receiving mentions from well-cited papers, by having central dependents, or by a combination of both.

We additionally provide analysis for three variations of our network: unweighted, weighted, and the largest connected component from the weighted graph. The unweighted form of our network ignores the weights added to the edges between the seeding articles and their software mentions. The weighted network includes the article citation count weights. The differences between the unweighted and weighted versions of our analysis allow us to parse different subtypes of the results. Specifically, comparing the weighted and unweighted networks allows us to identify software which is only important due to its utilization and reference in high citation count articles.

## 3 Results

In Fig 2, we show the overall network of software packages connected by their dependencies within each ecosystem and interconnected through the papers that mention them. Edges from papers to software packages are directed and weighted by the number of citations the paper received. Edges between software dependencies are directed from a dependent to a dependency.

We found a dense core of popular packages that receive many mentions (e.g., `ggplot2` in CRAN [33], `tophat` in PyPI and `limma` in Bioconductor [40,41]), some of which have many dependencies themselves (e.g., `ggplot2` [33]). We also found packages specific to certain communities (e.g., `PRISMA` in CRAN [35] or `pymol` in PyPI [37]).

Analyzing the entire ecosystem of software dependencies, we found that roughly 10% of software packages are part of dependency loops (i.e., cycles in the dependency networks). Interestingly, we found no cycles in the connected components of networks that have received software mentions, and their dependencies. We discuss this finding below. Note that contemporary package managers usually resolve the dependency loops by installing all packages in the loop simultaneously, so the loops, while indicating rather sloppy software practices, are not fatal.

In Fig 3, we show the distribution of packages from the three investigated ecosystems over Katz centrality and mention counts. We used three different ways to calculate centrality: (a) unweighted graph, (b) weighted complete graph, and (c) the largest connected components of the weighted graph (calculated separately for each ecosystem). It is remarkable that all three models give rather similar results for many packages. In the next section, we discuss the insights from the cases

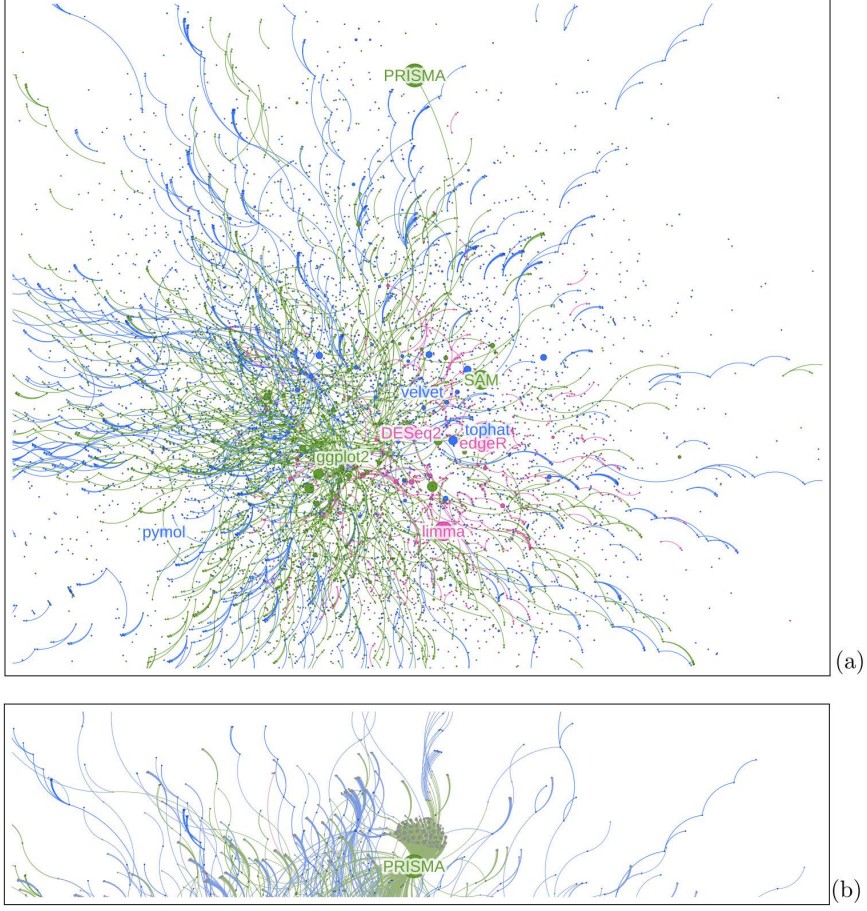

**Fig 2. (a) Network visualization of software packages from three ecosystems (from CRAN in green, PyPI in blue, and Bioconductor in pink) connected through their dependencies within their ecosystem and interconnected through papers that mention them.** We label the top 3 most central packages in each ecosystem: `ggplot2` [33], `SAM` [34], and `PRISMA` [35] for CRAN, `velvet` [36], `tophat` and `pymol` [37] for PyPI and `DESeq2` [38], `edgeR` [39] and `limma` [40,41] for Bioconductor. The core of the network is dominated by CRAN and PyPI dependencies, despite the fact that three of the five most central packages come from Bioconductor. (b) The top part of the above network, with papers added (in grey) to illustrate how `PRISMA` [35] can be central due to many mentions in papers.

when the different methods do not agree. As expected, we found a dense cluster of packages with low Katz centrality and low mention counts, i.e., the "Majority" quadrant (see Fig 1).

Fig 3 suggests large differences between the packages in the majority quadrant and the packages in the rest of the system. This is further illustrated by Tables 1 and 2. The sum of the median and half of the interquartile range is several orders of magnitude smaller than the maximal value for both the number of mentions and centrality.

A convenient measure of this inequality is the Gini coefficient [42]. Gini coefficients close to one characterize a very unequal distribution, while Gini coefficients close to zero correspond to a uniform distribution. As seen from Table 1, the distribution of mention counts is wildly unequal with high Gini coefficients. The distribution of centrality (Table 2) is smoother, especially for the weighted graph, but is still rather unequal.

Perhaps an even more robust measure of "Nebraska" properties of the ecosystem is the number of packages that are not mentioned in any paper, but are in the dependencies of the packages that are mentioned (we are grateful to the anonymous reviewer of PLOS Computational Biology for suggesting this metric). It is shown in Table 1.

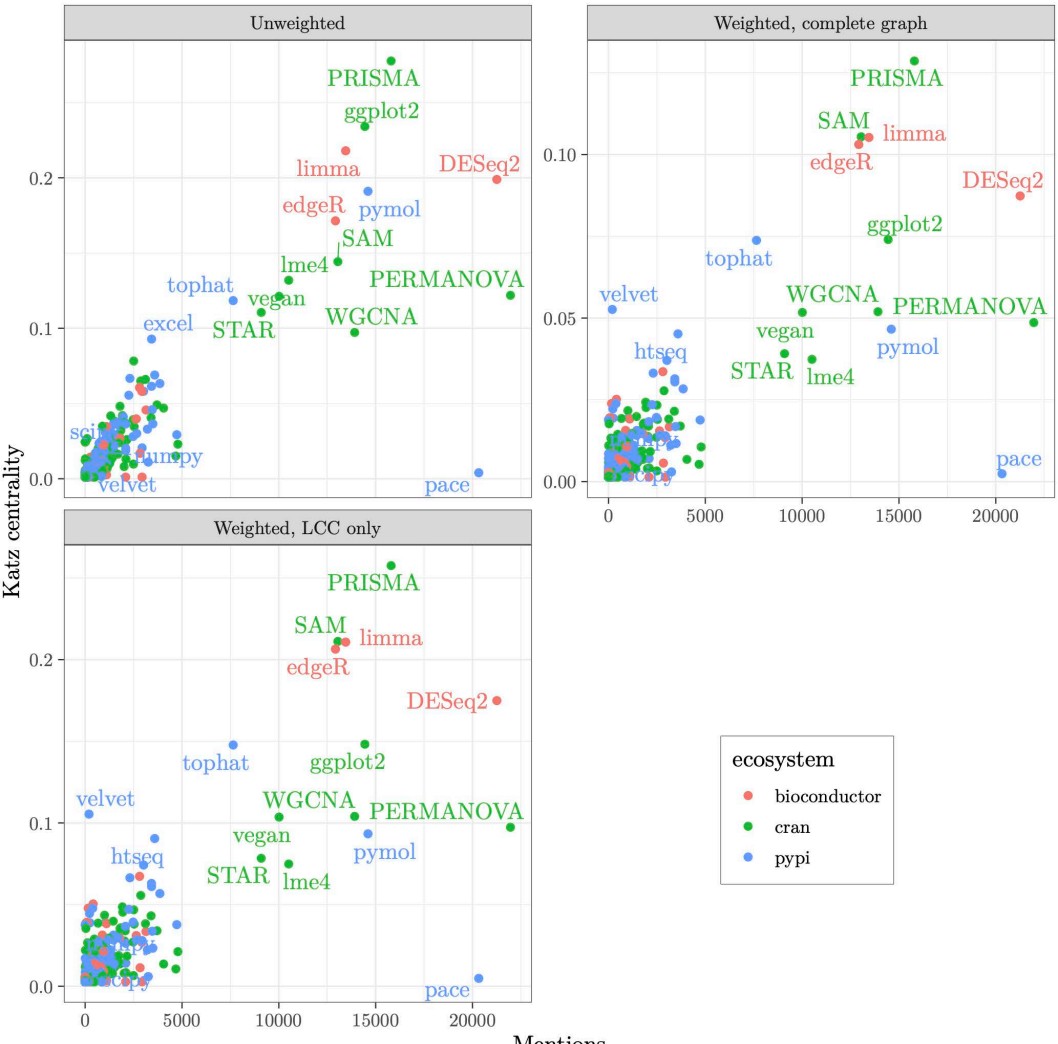

**Fig 3. Distribution of packages by Katz centrality and counts of their mentions in papers.** Katz centrality is calculated for an unweighted graph, for a weighted graph with all nodes, or just for the largest connected cluster (LCC) for each ecosystem. In the calculations, we assumed $\beta = 1$.

**Table 1. Summary statistics for the number of mentions in papers.** "Number" is the number of packages, "Dependency only" is the fraction of packages not mentioned in any papers and pulled by the dependencies only, "Median" is the median number of mentions, "IQR" is the inter-quartile range for mentions count, "Max" is the maximal number of mentions, and "Gini" is the Gini coefficient for the mentions count.

| Ecosystem | Number | Dependency only | Mentions count | | | |
|---|---|---|---|---|---|---|
| | | | Median | IQR | Max | Gini |
| bioconductor | 1018 | 0.047 | 24 | 68 | 21257 | 0.843712143943309 |
| cran | 3594 | 0.11324429604897 | 9 | 34 | 21960 | 0.889196914901722 |
| pypi | 5596 | 0.14349535382416 | 4 | 17 | 20322 | 0.888156736619396 |

**Table 2. Summary statistics for Katz centrality.** "Median" is the median centrality, "IQR" is the interquartile range, "Max" is the maximal centrality, and "Gini" is the Gini coefficient.

| Ecosystem | Median | IQR | Max | Gini |
|---|---|---|---|---|
| *Unweighted graph* | | | | |
| bioconductor | 0.001 | 0.00065127467783916 | 0.217934004136626 | 0.549779090129221 |
| cran | 0.00116646210956267 | 0.00048602587898445 | 0.277617982075916 | 0.496401393691949 |
| pypi | 0.00108092155486141 | 0.00019441035159378 | 0.191008170440888 | 0.351314534735671 |
| *Weighted, complete graph* | | | | |
| bioconductor | 0.00143155318286753 | 0.000322302811767757 | 0.105238137865536 | 0.358203102844196 |
| cran | 0.00138817278338669 | 0.000147428626544985 | 0.128618817548222 | 0.253046882585352 |
| pypi | 0.00138275023345159 | 0.00010438408625076 | 0.07377243622961 | 0.180193304141016 |
| *Weighted, LCC only* | | | | |
| bioconductor | 0.00288301570993583 | 0.00075920092370992 | 0.210742476047381 | 0.371663779294751 |
| cran | 0.00279343047601127 | 0.000369917791154937 | 0.257562976941226 | 0.280390754327007 |
| pypi | 0.00278528636383631 | 0.00031286964272138 | 0.147731480149716 | 0.220244615942456 |

In Tables 3, 4, and 5, we show the packages in "Pasteur", Popular, and "Nebraska" packages using the following criteria:

1. "Pasteur": the highest sum of mentions and centrality percentile. To provide a "sanity check", we added the data for three important packages: numpy, scipy, and napari.

2. Popular: the highest mentions percentile while centrality percentile is less than 0.5.

3. "Nebraska": the highest centrality percentile while mentions percentile is less than 0.5.

These metrics calculated for all packages are available as CSV files in the repository [22] (https://github.com/borisveytsman/SoftwareImpactHackathon2023_Tracing_dependencies/tree/main/data).

## 4 Discussion

Generally, the results verify our intuition that the majority of software packages will have a low Katz centrality and relatively few mentions. The outliers are then the most interesting packages, because they are central and often mentioned ("Pasteur"), or because their centrality outweighs their mentions ("Nebraska"), or vice versa (popular packages).

The absence of cycles in the network of open-source software packages used in our sample of biomedical science is quite interesting. It suggests a more robust design structure in the universe of scientific software than in the general software world. The lack of cycles also improves our analysis with Katz centrality, as the absence of loops excludes feedback mechanisms that can artificially inflate the centrality of packages on a loop. Packages with low mentions and high Katz centrality are therefore even more critical, as they represent essential dependencies of packages that enable large volumes of research without receiving direct attention. These are the "Nebraskan" packages referenced by [15] that we set out to find. Fig 3 and Tables 3–5 provide examples of these packages. As expected, most of these software packages are libraries used by other packages: vctrs [43] (a package for type manipulation in R), withr [44] (a set of tools for safely running functions that change global variables), isoband [45] (a package to generate isolines from gridded data), pauvre [46] (a package for plotting Oxford Nanopore data), newick [47] (read and write Newick data formatted files), setuptools [48] (a packaging library for Python projects). It is interesting that the prominent R "Nebraska" packages belong to the tidyverse system [49], and thus are co-authored by Hadley Wickham.

**Table 3. Software quadrants, unweighted graph.**

| Software | Ecosystem | Mentions | | Centrality | |
|---|---|---|---|---|---|
| | | Count | Percentile | Value | Percentile |
| **"Pasteur"** | | | | | |
| PRISMA | cran | 15797 | 0.9997 | 0.27762 | 1 |
| DESeq2 | bioconductor | 21257 | 0.9999 | 0.19888 | 0.9997 |
| ggplot2 | cran | 14441 | 0.9995 | 0.23407 | 0.9999 |
| pymol | pypi | 14604 | 0.9996 | 0.19101 | 0.9996 |
| PERMANOVA | cran | 21960 | 1 | 0.1219 | 0.9992 |
| limma | bioconductor | 13451 | 0.9993 | 0.21793 | 0.9998 |
| SAM | cran | 13048 | 0.9992 | 0.14425 | 0.9994 |
| edgeR | bioconductor | 12923 | 0.9991 | 0.17137 | 0.9995 |
| lme4 | cran | 10510 | 0.999 | 0.13191 | 0.9993 |
| WGCNA | cran | 13915 | 0.9994 | 0.09721 | 0.9988 |
| vegan | cran | 10012 | 0.9989 | 0.12121 | 0.9991 |
| tophat | pypi | 7642 | 0.9987 | 0.1184 | 0.999 |
| STAR | cran | 9091 | 0.9988 | 0.11052 | 0.9989 |
| .......... | | | | | |
| scipy | pypi | 1318 | 0.9921 | 0.02081 | 0.9916 |
| numpy | pypi | 962 | 0.9884 | 0.0176 | 0.99 |
| .......... | | | | | |
| napari | pypi | 11 | 0.5873 | 0.00118 | 0.6104 |
| **Popular** | | | | | |
| GSVA | bioconductor | 2937 | 0.9969 | 0.00108 | 0.361 |
| MAST | bioconductor | 2091 | 0.995 | 0.00108 | 0.361 |
| FSA | cran | 499 | 0.9782 | 0.00108 | 0.361 |
| Boruta | cran | 441 | 0.9754 | 0.00108 | 0.361 |
| MixSIAR | cran | 377 | 0.9714 | 0.00117 | 0.4989 |
| DRIMSeq | bioconductor | 335 | 0.9678 | 0.00117 | 0.4989 |
| netDx | bioconductor | 174 | 0.9419 | 0.00107 | 0 |
| COCOA | bioconductor | 162 | 0.9378 | 0.00117 | 0.4989 |
| emoji | pypi | 161 | 0.9374 | 0.00107 | 0 |
| aviso | pypi | 160 | 0.937 | 0.00117 | 0.4989 |
| SiZer | cran | 159 | 0.9366 | 0.00117 | 0.4989 |
| searchlight | pypi | 137 | 0.9273 | 0.00107 | 0 |
| **"Nebraska"** | | | | | |
| vctrs | cran | 2 | 0.2516 | 0.02438 | 0.9928 |
| withr | cran | 2 | 0.2516 | 0.02438 | 0.9928 |
| isoband | cran | 0 | 0 | 0.02438 | 0.9928 |
| gss | cran | 7 | 0.4991 | 0.01212 | 0.9849 |
| SuppDists | cran | 3 | 0.3287 | 0.00567 | 0.9658 |
| lfda | cran | 1 | 0.1232 | 0.0049 | 0.9581 |
| ggtext | cran | 7 | 0.4991 | 0.00482 | 0.957 |
| texttable | pypi | 1 | 0.1232 | 0.00478 | 0.9569 |
| affyio | bioconductor | 5 | 0.4326 | 0.00473 | 0.956 |
| dnaio | pypi | 1 | 0.1232 | 0.00472 | 0.9558 |
| requests | pypi | 7 | 0.4991 | 0.00419 | 0.9499 |
| tifffile | pypi | 4 | 0.3863 | 0.00409 | 0.9488 |

**Table 4. Software quadrants, weighted graph (full).**

| Software | Ecosystem | Mentions | | Centrality | |
|---|---|---|---|---|---|
| | | Count | Percentile | Value | Percentile |
| **"Pasteur"** | | | | | |
| PRISMA | cran | 15797 | 0.9997 | 0.12862 | 1 |
| DESeq2 | bioconductor | 21257 | 0.9999 | 0.08735 | 0.9996 |
| SAM | cran | 13048 | 0.9992 | 0.10545 | 0.9999 |
| limma | bioconductor | 13451 | 0.9993 | 0.10524 | 0.9998 |
| ggplot2 | cran | 14441 | 0.9995 | 0.07401 | 0.9995 |
| PERMANOVA | cran | 21960 | 1 | 0.04861 | 0.999 |
| edgeR | bioconductor | 12923 | 0.9991 | 0.10309 | 0.9997 |
| WGCNA | cran | 13915 | 0.9994 | 0.05195 | 0.9992 |
| pymol | pypi | 14604 | 0.9996 | 0.04661 | 0.9989 |
| tophat | pypi | 7642 | 0.9987 | 0.07377 | 0.9994 |
| vegan | cran | 10012 | 0.9989 | 0.05172 | 0.9991 |
| lme4 | cran | 10510 | 0.999 | 0.03738 | 0.9986 |
| .......... | | | | | |
| scipy | pypi | 1318 | 0.9921 | 0.00762 | 0.9881 |
| numpy | pypi | 962 | 0.9884 | 0.0083 | 0.9891 |
| .......... | | | | | |
| napari | pypi | 11 | 0.5873 | 0.00136 | 0.2967 |
| **Popular** | | | | | |
| GSVA | bioconductor | 2937 | 0.9969 | 0.00136 | 0.1214 |
| MAST | bioconductor | 2091 | 0.995 | 0.00136 | 0.1214 |
| GSA | cran | 844 | 0.9868 | 0.00136 | 0.2163 |
| FSA | cran | 499 | 0.9782 | 0.00136 | 0.1508 |
| Boruta | cran | 441 | 0.9754 | 0.00136 | 0.0968 |
| MixSIAR | cran | 377 | 0.9714 | 0.00138 | 0.4265 |
| AntWeb | cran | 282 | 0.9616 | 0.00138 | 0.4265 |
| emoji | pypi | 161 | 0.9374 | 0.00138 | 0.4356 |
| eureqa | pypi | 149 | 0.9331 | 0.00138 | 0.4265 |
| searchlight | pypi | 137 | 0.9273 | 0.00136 | 0.259 |
| pathfindR | cran | 133 | 0.9259 | 0.00137 | 0.4023 |
| ChIPXpress | bioconductor | 125 | 0.9213 | 0.00138 | 0.4129 |
| pypet | pypi | 125 | 0.9213 | 0.00136 | 0 |
| ADAPTS | cran | 125 | 0.9213 | 0.00137 | 0.3452 |
| **"Nebraska"** | | | | | |
| calculate_expression | pypi | 3 | 0.3287 | 0.01872 | 0.9956 |
| zscore | pypi | 5 | 0.4326 | 0.00851 | 0.9893 |
| webpages | pypi | 2 | 0.2516 | 0.00707 | 0.9872 |
| setuptools | pypi | 6 | 0.4689 | 0.00572 | 0.9825 |
| gpviz | pypi | 1 | 0.1232 | 0.00453 | 0.9759 |
| glib | pypi | 3 | 0.3287 | 0.00397 | 0.9711 |
| ego | pypi | 5 | 0.4326 | 0.00326 | 0.9607 |
| xtick | pypi | 1 | 0.1232 | 0.00309 | 0.9575 |
| pauvre | pypi | 5 | 0.4326 | 0.00292 | 0.9531 |
| mappy | pypi | 4 | 0.3863 | 0.00292 | 0.9531 |
| rle | cran | 2 | 0.2516 | 0.00287 | 0.9522 |
| LineagePulse | bioconductor | 1 | 0.1232 | 0.0027 | 0.9474 |

**Table 5. Software quadrants, weighted graph (LCC component only).**

| Software | Ecosystem | Mentions | | Centrality | |
|---|---|---|---|---|---|
| | | Count | Percentile | Value | Percentile |
| **"Pasteur"** | | | | | |
| PRISMA | cran | 15797 | 0.9996 | 0.25756 | 1 |
| DESeq2 | bioconductor | 21257 | 0.9999 | 0.17492 | 0.9995 |
| SAM | cran | 13048 | 0.999 | 0.21116 | 0.9999 |
| limma | bioconductor | 13451 | 0.9991 | 0.21074 | 0.9998 |
| ggplot2 | cran | 14441 | 0.9994 | 0.1482 | 0.9994 |
| PERMANOVA | cran | 21960 | 1 | 0.09735 | 0.9988 |
| edgeR | bioconductor | 12923 | 0.9989 | 0.20643 | 0.9996 |
| WGCNA | cran | 13915 | 0.9993 | 0.10402 | 0.999 |
| pymol | pypi | 14604 | 0.9995 | 0.09333 | 0.9986 |
| tophat | pypi | 7642 | 0.9984 | 0.14773 | 0.9993 |
| vegan | cran | 10012 | 0.9986 | 0.10357 | 0.9989 |
| lme4 | cran | 10510 | 0.9988 | 0.07485 | 0.9983 |
| .......... | | | | | |
| scipy | pypi | 1318 | 0.99 | 0.01525 | 0.9851 |
| numpy | pypi | 962 | 0.9854 | 0.01662 | 0.9863 |
| .......... | | | | | |
| napari | pypi | 11 | 0.5351 | 0.00273 | 0.2816 |
| **Popular** | | | | | |
| GSVA | bioconductor | 2937 | 0.996 | 0.00272 | 0.1185 |
| MAST | bioconductor | 2091 | 0.9937 | 0.00272 | 0.1185 |
| GSA | cran | 844 | 0.9833 | 0.00272 | 0.2167 |
| FSA | cran | 499 | 0.9724 | 0.00272 | 0.148 |
| Boruta | cran | 441 | 0.969 | 0.00272 | 0.0923 |
| MixSIAR | cran | 377 | 0.9639 | 0.00276 | 0.3878 |
| AntWeb | cran | 282 | 0.9516 | 0.00276 | 0.3878 |
| emoji | pypi | 161 | 0.9215 | 0.00276 | 0.397 |
| searchlight | pypi | 137 | 0.9091 | 0.00273 | 0.254 |
| pathfindR | cran | 133 | 0.9073 | 0.00275 | 0.3677 |
| ChIPXpress | bioconductor | 125 | 0.9018 | 0.00276 | 0.3765 |
| pypet | pypi | 125 | 0.9018 | 0.00272 | 0 |
| **"Nebraska"** | | | | | |
| calculate_expression | pypi | 3 | 0.2982 | 0.0375 | 0.9944 |
| zscore | pypi | 5 | 0.3883 | 0.01705 | 0.9865 |
| setuptools | pypi | 6 | 0.4215 | 0.01146 | 0.978 |
| newick | pypi | 9 | 0.4957 | 0.00806 | 0.9649 |
| xtick | pypi | 1 | 0.1305 | 0.00618 | 0.9469 |
| nanolyse | pypi | 9 | 0.4957 | 0.00595 | 0.9443 |
| splatter | pypi | 8 | 0.4742 | 0.00584 | 0.9422 |
| pauvre | pypi | 5 | 0.3883 | 0.00584 | 0.9413 |
| mappy | pypi | 4 | 0.3474 | 0.00584 | 0.9413 |
| rle | cran | 2 | 0.232 | 0.00574 | 0.9402 |
| scone | pypi | 8 | 0.4742 | 0.00574 | 0.9398 |
| LineagePulse | bioconductor | 1 | 0.1305 | 0.0054 | 0.9344 |

To check the results, we included in the tables three popular Python packages: `scipy` [50], `numpy` [51], and `napari` [52]. We see that `scipy` and `numpy` are in high percentiles of package by centrality and mentions for all three versions of network studied. The newer and more specialized package `napari` is lower in the ratings, but still is above 50% percentile. All this seems reasonable.

Similarly we were pleased to note that the analysis identified the `tifffile` [53] package as part of the Nebraska quadrant and quick inspection of the online materials confirm this as a high quality, widely used package with one person development team, albeit in California and not physically in Nebraska.

The differences between the unweighted and weighted versions of our analysis, shown in Fig 3, can help us parse different subtypes of these outliers. For example, the `velvet` [54] package is central solely because it is mentioned by one highly cited paper (and dozens of less highly cited ones). Such characteristics do not make a package central in the unweighted version of the analysis, since citation counts are ignored there. Central packages under this representation, therefore, also need to have other packages depend on them and cannot solely rely on mentions from influential papers.

It is interesting that `cran` and `bioconductor` dominate the "Nebraska" quadrant for the unweighted graph, while for the weighted graph, `pypi` is overrepresented. The reason for this difference may warrant additional research.

Altogether, our paper tries to quantify impact as importance and to provide a set of metrics behind the intuition. Of course, this is by itself a perilous process: there could be various flavors of both impact and importance, and people may reasonably disagree about the details. Our contribution is to propose a certain quantification, which agrees with our intuitive understanding.

In addition to possible problems with the exact definition of importance, our work has a number of limitations.

1. The network analysis is limited by the quality of disambiguation and linking in the CZ Software Mentions dataset (see the discussion in [10]). Homonyms (different software packages with the same or close names) may change the network statistics, and preliminary research points to considerable potential for incorrect linking in the CZ Software Mentions dataset due to homonyms [6]. Notably, this affects the packages `velvet` and `tophat` (see Fig 2a), which have been falsely linked to Python packages. `velvet` was linked to a package that provides signal processing and communications algorithms in Python [36]. Meanwhile, *all* of the mentions of "velvet" point to different versions of algorithms for de novo short-read assembly using de Bruijn graphs in genomics, as described in [55] and implemented mostly in C [54]. `tophat` was linked to a package that provides a framework for collaborative and multiplayer mobile applications [56]. However, over 99% of mentions point to different software called `TopHat`, a splice junction mapper for RNA-Seq reads, implemented mostly in C**++** [57,58], and the remainder to the Top-Hat transform algorithm used for baseline subtraction in image processing. While the linking errors do not change the centrality values and our interpretation of them – both rely on disambiguation, not linking – they led to mislabeling of different, mostly non-PyPI packages as PyPI packages for the visualization in Fig 2. Sometimes an open source package also shares a name with a proprietary software (e.g., PRISM, PACE), which leads to confusion and incorrect mention counts. Nonetheless, we think that this contribution is useful, both as a demonstration of this approach and because our analysis is reproducible. This means that limitations can be addressed in improvements over time. For example, as datasets emerge with improvements to disambiguation these datasets can be incorporated into the analysis and comparisons or updates produced.

2. Our approach is limited to the open-source packages available in one of the chosen ecosystems. This excludes proprietary software like Excel or GraphPad Prism [59], and open-source non-package software like Gephi [60] (a compiled application). In addition, our work excludes software projects such as BLAS [61], LAPACK [62] or the Linux Kernel itself. These packages are critically important but are often lower-level system dependencies that may not be clearly cited or referenced in an article's text or provided in a software library's dependency list. Without such references or dependencies, it is hard to detect their specific utilization. Without authors including references to lower-level software libraries in their articles, and without explicit dependency on lower-level libraries from mentioned software, we simply

cannot include such software in our networks. The number of missing packages from the ecosystems investigated may be estimated from Table 14 of [10], which estimates the coverage for GitHub at 64.39%, the coverage for CRAN at 8.36%, the coverage for PyPI at 5.86%, and the coverage for Bioconductor at 3.23%. GitHub coverage can be used as a proxy for the share of the open source software: while some packages—perhaps growing in number—are not hosted by GitHub, their fraction remains low [51]. If we assume that each package exists only in one ecosystem (there are packages present in several, but we neglect this effect), the combined coverage by CRAN, PyPI, and Bioconductor is 17.45%, i.e., between a quarter and one third of the open source packages. It is very likely that each of these non-package pieces of software do depend on packages in our focus ecosystems. For open-source non-package software, it may be feasible to identify source repository URLs and then identify packages depended on by the software. That would need to be done by directly inspecting the source code, possibly using Software Bill of Materials (SBOM) tools [12]. On the other hand, some non-package software does not have source code available (including proprietary GUI software such as SPSS, and cloud-based services). Nonetheless, these pieces of code may very well be based on packages in our focal ecosystem (especially if the latter use non-restrictive licenses such as BSD, MIT). In the longer term, there is a possibility that the requirement to provide Software Bills of Materials (SBOMs) for U.S. federal government purchasers will give insight into the packages on which proprietary software relies [63].

3. In our dependency network, we always used the latest version of any package. In reality, dependencies change between the versions. A study of development logs often reveals messages like "deleted the dependence on XYZ" or "added the dependence on ABC". These changes often make the work of package management software very difficult. A software dependency graph is a living network, constantly evolving, with links added and deleted. In this study, we effectively collapse the time, making an (imperfect) snapshot of the long movie. A future improvement would be to use each package ecosystem's specific dependency resolution algorithm to compute the full transitive dependency tree for each mentioned software package. An even further extension would be to attempt to do this for the version of the package most likely used by a specific publication, based on aligning publication date with the current version used at that time.

4. Our method captures the off-the-shelf software packages used for each article. It does not capture *ad hoc* software written specifically for the given paper, which may be contained in the code accompanying the papers. This code may load software libraries and thus add software dependencies that are absent from our graph.

5. The dependencies reflected in the package metadata do not necessarily reflect the actual usage of the software. A library may be loaded, but not used for the actual execution by a given paper. A more reliable method would be to reproduce the analysis done in each paper and capture the actual library calls—which is probably prohibitively difficult for the number of papers covered.

6. Many Python and R packages ultimately depend on libraries outside their ecosystems (most often C and Fortran libraries). These dependencies are not easily captured by the package management infrastructure and thus are outside of our analysis.

7. Our dependency network does not include the concept of alternatives, when package $A$ depends on the functionality that can be provided either by package $B_1$, or package $B_2$. Some package managers like Debian's *apt* have the possibility of specifying alternatives, but this is not a common feature of package managers.

8. Different communities place different emphasis on infrastructure elements such as tests. This may lead to the over-representation of testing infrastructure packages in some cases, for example, in the Python ecosystem. We do not try to attempt to decide which elements of the infrastructure are "actually important"; instead, we rely on the judgment of the community of users and developers while recognizing that such judgment may be different across the landscape.

9. Our current approach does not filter or subset the network into different time periods (i.e., by year) and as such, some of the software identified as central to our network such as `velvet` and `tophat` are perhaps historical artifacts that have not been updated in many years. These examples demonstrate that future analysis of software dependency networks should consider temporal data to understand how particular software projects (and more broadly, computational methods) rise in adoption and then decline as new methods become available.

10. There are different software package repositories and ecosystems for Python and R with greatly overlapping dependency graphs. We used PyPI, CRAN and Bioconductor for this study. Alternative repositories that provide packages for Python and R, such as Conda channels (conda-forge, bioconda, etc.), are also worth investigating as there may be some software projects distributed solely through Conda channels. Conda provides alternate packaging standards from PyPI and CRAN and as such, over a large distribution, the same set of packages sourced from PyPI may have a slightly different dependency graph than one sourced from Conda. Conda also has more cross-language dependencies listed and may allow deeper traversal to find infrastructural libraries like BLAS; other improvements could include using PyPI Wheels or HomeBrew dependency systems.

## 5 Conclusions and future work

In this work, we investigated the network of software packages used in the biomedical literature available through PubMed Central. We demonstrated that these packages follow the structure of "Stokes' diagram" (famous for "Pasteur's quadrant"), with some packages highly visible to the end users, and some packages less visible, but important in the network due to their dependencies. We discussed the use of Katz centrality in discovering the important packages and found examples of such packages in the biomedical field.

These findings and insights, as well as the underlying methodology, can be used to identify critical, but low-visibility, open-source scientific software in need of targeted funding and support.

Of course, there are many ways to extend this work. It would be interesting, for instance, to analyze common workflows for different disciplines, perhaps using co-occurrences of mentions, and map them into the dependency network. This might help to discover packages important for specific sub-fields of biomedical sciences. Similarly, adding temporal dependencies to our graph may help to discover and predict development trends. Additionally, individual dependencies may differ in the actual impact that they have on the dependent software (i.e., a dependency that is only minimally utilized by the downstream software itself may be given less weight). Developing and applying a metric that describes these differences in impact would make it possible to refine our approach to quantifying the importance of research software dependencies.

## Author contributions

**Conceptualization:** Stephan Druskat, James Howison, Daniel Mietchen, João Felipe Pimentel, Boris Veytsman.

**Data curation:** Eva Maxfield Brown, Andrew Nesbitt, Boris Veytsman.

**Formal analysis:** Eva Maxfield Brown, Laurent Hébert-Dufresne, Andrew Nesbitt, Boris Veytsman.

**Investigation:** Eva Maxfield Brown, Stephan Druskat, Laurent Hébert-Dufresne, Andrew Nesbitt, Boris Veytsman.

**Methodology:** Eva Maxfield Brown, Stephan Druskat, Laurent Hébert-Dufresne, Andrew Nesbitt, Boris Veytsman.

**Software:** Eva Maxfield Brown, Stephan Druskat, Andrew Nesbitt.

**Validation:** Stephan Druskat.

**Visualization:** Eva Maxfield Brown, Laurent Hébert-Dufresne, Andrew Nesbitt, Boris Veytsman.

**Writing – original draft:** Eva Maxfield Brown, Stephan Druskat, Laurent Hébert-Dufresne, Andrew Nesbitt, Boris Veytsman.

**Writing – review & editing:** Stephan Druskat, James Howison, Daniel Mietchen, João Felipe Pimentel, Boris Veytsman.

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
