## [Decision Letter · Decision Letter 0]

4 Feb 2025

PCOMPBIOL-D-24-02238

Biomedical Open Source Software: Crucial Packages and Hidden Heroes

PLOS Computational Biology

Dear Dr. Veytsman,

Thank you for submitting your manuscript to PLOS Computational Biology. After careful consideration, we feel that it has merit but does not fully meet PLOS Computational Biology's publication criteria as it currently stands. Therefore, we invite you to submit a revised version of the manuscript that addresses the points raised during the review process.

Please submit your revised manuscript within 60 days Apr 06 2025 11:59PM. If you will need more time than this to complete your revisions, please reply to this message or contact the journal office at ploscompbiol@plos.org. Please include the following items when submitting your revised manuscript:

We look forward to receiving your revised manuscript.

Kind regards,

Varun Dutt, Ph.D

Academic Editor

PLOS Computational Biology

Mark Alber

Section Editor

PLOS Computational Biology

**Journal Requirements:**

At this stage, the following Authors/Authors require contributions: Andrew Nesbitt, Boris A Veytsman, Daniel Mietchen, Eva Maxfield Brown, James Howison, João Felipe Pimentel, Laurent Hébert-Dufresne, and Stephan Druskat. Please ensure that the full contributions of each author are acknowledged in the "Add/Edit/Remove Authors" section of our submission form.

6) Your current Financial Disclosure states, "The authors are grateful to the Chan Zuckerberg Initiative for making this re-search possible and for sponsoring the Mapping the Impact of Research Softwarein Science hackathon. L.H.-D. acknowledges support from the National Insti-tute of General Medical Sciences under the 2P20GM125498 Centers of Biomed-ical Research Excellence Award and from Google Open Source through theOpen-Source Complex Ecosystems And Networks (OCEAN) project. D.M. ac-knowledges support from the German Research Council (DFG) through theMaRDI project (460135501) as well as through the BASE4NFDI/KGI4NFDIproject (521453681). J.F.P. acknowledges support from CNPq and FAPERJ(E-26/210.478/2024)."

However, your funding information on the submission form does not indicate any funds. Please ensure that the funders and grant numbers match between the Financial Disclosure field and the Funding Information tab in your submission form. Note that the funders must be provided in the same order in both places as well.

Please indicate by return email the full and correct funding information for your study and confirm the order in which funding contributions should appear. Please be sure to indicate whether the funders played any role in the study design, data collection and analysis, decision to publish, or preparation of the manuscript.

**Reviewers' comments:**

Reviewer's Responses to Questions

Reviewer #1: This manuscript by Nesbitt et al. presents a study of the graph of citations between papers and software tools as well as between software tools themselves. The main thesis put forward is that the Katz centrality metric is a good way to measure the importance of papers and software tools and can capture tools that are under-appreciated. While this is interesting, the manuscript is lacking in important details and, in my opinion, fails to provide a convincing argument for this thesis.

1. The authors justify the use of the Katz centrality metric as fulfilling some basic criteria, but other than excluding two related metrics (eigencentrality and PageRank) the argument for Katz centrality is not very well developed, neither from a theoretical (e.g., deriving from a set of axioms) or from a purely empirical perspective.

2. The authors write that "We found that Katz centrality [Katz, 1953] satisfies all our criteria. This approach gives us the attenuation factor β as a free parameter to control the importance of papers..." This requires more detail. The original [Katz, 1953] paper does not mention a parameter β (in fact, it is typeset without the use of Greek letters, and mentions only the attenuation factor a, which today we would typeset as α). The authors should clarify what they mean by this and present the actual formulas they use. Is β the attenuation factor? It does not seem to be the case as it would not be related to the importance of papers (at least in my understanding, but I may be wrong given the lack of detail in the manuscript).

3. The authors acknowledge that the "velvet" is a misassignment. The velvet assembler is hardly under-appreciated (google scholar gives it >11,000 citations: https://genome.cshlp.org/content/18/5/821.short, https://scholar.google.com/scholar?cites=355244555212756650). Having had a look at their dataset, it seems that the highly cited paper mentioned in the text appears to be 10.1093/bioinformatics/btu170 (the Trimmomatic paper) which is marked as citing "velvet", but reading that paper, it cites the velvet assembler, not the velvet package on PyPI.

Without this example, there are no other examples of under-appreciated tools in the manuscript, which was a centrally motivating factor for the study. In fact, from Figure 3, I would even be tempted to say that citations do a good job of capturing the centrality of tools, although it is also difficult to say because this figure does not provide a lot of detail for the papers with <5000 citations (as the dots overlap and are impossible to distinguish), which constitute the majority of papers.

4. The code is available, but references paths such as `~/Downloads/pruned-network.csv` which are not directly usable.

Reviewer #2: Open-source software or libraries used in a scientific context are notoriously seldom cited, if they are cited at all. As highlighted by the authors, the problem of identifying packages crucial to the proper functioning of scientific infrastructure appears even more challenging when considering the breadth of packages used as dependencies hidden to the users of scientific software. How can we, then, as scientists using software everyday, support critical software packages when they cannot be identified so easily? In their manuscript, the authors proposed a method to identify such critical packages by merging information from a large public dataset of software mentions in the biomedical literature (CZI Software Mentions Dataset) and the dependencies trees of the mentioned software into a single network. Because this dataset was born out of data extracted from biomedical papers, their network has the potential to be representative of the open-source biomedical software infrastructure.

While this work has a lot of potential, the manuscript suffers from a serious lack of tangible findings that would confirm the validity of the approach. The results consist in a dense graph from which users cannot draw conclusions, and the name of 9-16 packages identified as the most “central” given the authors’ metrics. There is little to no investigation of these packages, which are in any case far too few to allow the authors to conclude anything about the approach. The only package for which some scrutiny has been applied is held as a new-found critical package that is hardly mentioned in the literature (velvet), but the authors themselves cast shadows on the veracity of that finding since it only appears because its original method is historically important (see first major comment). The authors identify and clearly state many interesting limitations to their approach (which is laudable), whether from a consequence of imperfect data or of their metrics itself. But they do not attempt to quantify how much these limitations are impacting their measure through validation of the prominent packages or known/expected packages (see major comments 2 and 3). Finally, I find the manuscript difficult to parse and suggest several axis of improvements, as well as additional information that would make it easier for readers to judge the method.

In its current state, I do not think that the manuscript can be published. I do believe, however, that both the dataset and the approach have the potential to yield interesting results should informative data analysis be applied.

## Major comments

1. One of the main result (if not the only one that is actually discussed) of the paper is the identification of “velvet” as a “Nebraskan” package. Two aspects of that result bother me: firstly in the way it is introduced, and secondly in what it actually says about the method. In the discussion section, “velvet” is first introduced as the only (for lack of another one mentioned) “Nebraskan” package, and the authors then state that the package disappears as soon as the paper citation counts are not taken into account (unweighted graph). In another section of the discussion, they double down on by highlighting again that it is mostly, if not only, central because of one highly cited paper (presumably the original publication).

In my opinion, this totally undermines the method, which aims at identifying specifically “Nebraskan” packages. Indeed, the only identified “Nebraskan” package only pops in the analysis because the original method is highly cited (11000 citations)... While I cannot exclude that the package is still used (there are a few issues from 2022 in the Github repository), it clearly is not a dependency of other software (from the author’s own analysis), it is 17 years old (last commit 11 years ago) and the paper is probably cited because of its historical importance. A quick question to genomics specialist made it clear that newer tools are more useful nowadays: Abyss, PacBio, Nanopore. Genomics being such an active field, I have a hard time believing that velvet is still crucial to the scientific infrastructure.

To justify the power of the method, better (so far unknown) examples should be highlighted to the readership.

2. As exemplified by velvet, the manuscript lacks analysis validating the method. Only 9 (fig 2, or 16 in fig 3) packages are mentioned, of which a single one is investigated. I strongly suggest examining closely a non-negligible number of packages (e.g. 50-100, number should be adjusted based on how much work it ends up being, while maintaining enough statistics to be able to draw conclusions) that fall into the “Nebraskan” and “Pasteur” categories (given ad-hoc thresholds on mentions and centrality) and identifying the amount of “potential impostors” (e.g. like velvet) in that pool.

The authors should go through the list of identified packages and compile metrics and metadata information to validate whether the package could belong to the assigned category:

- Is it an homonym?

- Centrality in weighted vs unweighted graphs

- Number of mentions in the CZI dataset

- Download statistics (which are available in all three package ecosystems investigated)

- Average citation / paper with edges to this package, highest citation count for paper with edge (aka guarding against velvet-like packages)

- Date of the latest commit in Github / Date of the latest package version

- Rate of updates (e.g. average commits / months in the last year or versions / year in the last 5 years)

- Number of Github stars if applicable

- …

None of these are perfect proxy metrics (e.g. download statistics will not be comparable between ecosystems, CI/CD can skew the downloads etc.), but they can all inform to a certain extent on whether there might be issues with the particular software packages.

Additionally, assigning a software category to these packages would allow refining the analysis and help understand what type of packages the method is able to classify. I don’t know if ontologies already exist, but categories could be along the lines of: specific analysis (BLAST, velvet), generic computation (scikit-learn), infrastructure / data structures (numpy, pytest, pandas), rendering (ggplot2, matplotlib).

The results could be show in tables, for instance per categories and per ecosystem, before being discussed.

3. An additional important experiment to validate the method is to build a list of known packages and inspect where they appear in the [Mentions, Centrality] graph. While the authors correctly state that different fields will assign importance to different packages, and such importance will indeed reflect mostly direct use of package (as opposed to usage through dependencies), this does not mean that one should not investigate what happens to clearly important scientific software package. Such a list can easily be built by surveying colleagues or asking in forums (biostars and image.sc already cover a large range of biomedical applications, missing mainly the medical fields). The authors could make sure that they are building a list that covers examples in categories spanning application of a particular analysis, generic analysis package, data i/o, rendering or more infrastructure ones. Examples that come to mind are for instance dplyr (I am limited in my knowledge of R), xgboost, pandas, matplotlib, scipy, scikit-learn, dask, pytorch, ggplot2 (obviously that one is already clearly identified in the current analysis, although it is assumed that the readers are familiar with it and it is not discussed).

The centrality and mentions measures could be reported for these packages per ecosystem and discussed in comparisons to those identified via thresholding the metrics (Nebraskan and Pasteur), as well as some of the other metrics.

4. I find the manuscript difficult to read for multiple reasons:

- The Stokes-like diagram is central in the introduction, without describing what a “Pasteur” package is, forcing readers to google the Stokes diagram. “Nebraskan” and “Pasteur” are mentioned in the text a lot, without exemplifying it in the result figures.

- Mentioned software packages should be identified with references and links to the package as well as with a short description of what they are. Otherwise, readers are forced to investigate on their own what these packages are (e.g. I had to google velvet, and until I read the discussion I had no idea why this package was picked up by the analysis at all).

- Discussion about the packages are done piece-wise in multiple places and make it difficult to follow the logic regarding the package (velvet).

- A substantial part of the discussion is actually results (metrics for velvet, appearance and its disappearance in the different graphs, package coverage).

5. The authors state that they are using the CZI Software Mentions Dataset in the introduction, but do not describe it. Readers have to wait for the discussion to learn about potential pitfalls (cluster of names, homonyms) that may impede the analysis. In addition, the authors should add a few numbers to highlight the importance of the dataset: e.g. number of papers and number of software in the dataset.

6. As opposed to what is stated in multiple places (e.g. point 9 in the discussion) in the paper, PyPI is not a package manager, but a package repository (same for CRAN?). pip/conda are indeed package managers for the Python ecosystem.

7. First paragraph introduction: “This is true not only for the sciences”, prefer the term “natural sciences” to not appear derogatory to sociology.

8. Paragraph 2 and 4 in introduction repeat each other.

9. The authors state that “We also found less central packages mentioned heavily by specific communities (e.g. PRISMA in CRAN […]”, but it seems to me that PRISMA is the most central package identified by their method (Fig 3) solely based on the metrics.

10. The authors should mention in the result sections how many software packages are identified through the dependency trees, that is the total number of software packages in their graph.

11. The result section or the supplementary materials should provide distributions of centrality and of mentions, as well as averages. A comparison of how the distribution of centrality measures changes between unweighted and weighted would be interesting.

12. Could the authors discuss the largest connected component? How do they interpret it? How come it seems to be cross-ecosystem? If a paper mentions both python and R package, does that make their dependency tree a single connected component?

13. I tried running the Github example notebooks, but these failed due to json errors or missing json (e.g. pypi_with_mentions.ndjson). I have not investigated further. I would suggest making sure that the notebooks run, and that the README explains clearly how to get the data, run the analysis and explore the results in order to benefit the readership!

## Minor comments

1. It would be nice to add explanations about how Katz centrality is calculated in your analysis in the supplementary materials of the paper.

2. The authors noted that “we found that roughly 10% of software packages are part of dependency loops”. I am a bit confused by this, do they expect these to be working software? Can such software build in the R and Python world?

3. When the authors state that “It suggests a more robust design structure in the universe of scientific software than in the general software world.”, I would precise “open source software world” as company code / closed source software are often better organized due to internal constraints. There are other mentions of this general software world.

4. When the authors state that “The lack of cycles also improves our analysis with Katz centrality, as it excludes feedback mechanisms that can artificially inflate the centrality of packages on a loop”, do they mean that the lack of cycles exclude feedback mechanisms? This makes sense to me, but on the first read I had the impression that the Katz centrality calculation excluded feedback mechanisms.

5. It is hinted at, but worth insisting on for the readership that, at a time where research institutions have a hard time battling constant cybersecurity targeting, open source software are potential entry ways for malicious actors (similar to the XZ Utils case).

Reviewer #3: This paper takes steps to tackle an important and timely issue in bioinformatics, to credit the hidden labour of key software resources and to drive targeted sustainability efforts for key but invisible software packages. Software is the life blood of data driven biology research. Tractably building a credible evidence base in the complex network of modern software is fundamental to these goals.

The paper is well written and balanced, reporting an early effort using the CZI software mentions dataset and the Ecosyste.ms dataset. It is balanced because the datasets have a number of limitations, the analysis makes a number of assumptions and there are fundamental issues in the way software is packaged and deemed important that make this an exploratory exercise. The authors make these limitations explicit, effectively laying down a roadmap of challenges for future work and other researchers to take up.

Given that “perfect is the enemy of good” these challenges should not hold back the effort.

The network analysis and weighting choices are convincingly presented and chimes with intuition.

• It would have been interesting to approach leaders in the CRAN, PyPi and Bioconductor community for their views in the results - are there surprises or is it confirming intuition?

• The lack of dependency cycles is suggested as robust design - is it just less complexity or more homogeneity in the scientific software world? Or less legacy? Or smaller well connected development communities?

• Figure 3 the weighted complete graph and the LCC cluster pattern is identical (as could be expected) so it’s unclear what the point is of showing this is.

• Figure 2 - apart from the labelled packages the rest of the figure is tricky to interpret. Unless it is further discussed its value is not clear

• The paper focus was in three packages in each ecosystem. What about the next 10?

• The bioscience literature community is interesting - there are dedicated journals for bioinformatics and others for scientific discovery that may mention the immediate software. Are there any differences in the citation (and hence weighting) pattern between different classes of journal ?

As this is a first step a future work paragraph on how the community might prepare cleaner datasets and further analysis to address limitations of the work in order to fulfil the ultimate credit and funding goals

**Have the authors made all data and (if applicable) computational code underlying the findings in their manuscript fully available?**

Reviewer #1: Yes

Reviewer #2: Yes

Reviewer #3: Yes

PLOS authors have the option to publish the peer review history of their article (what does this mean?). If published, this will include your full peer review and any attached files.

Reviewer #1: No

Reviewer #2: No

Reviewer #3: No

**Figure resubmission:**
---

## [Decision Letter · Decision Letter 1]

14 Aug 2025

PCOMPBIOL-D-24-02238R1

Biomedical open source software: Crucial packages and hidden heroes

PLOS Computational Biology

Dear Dr. Howison,

Thank you for submitting your manuscript to PLOS Computational Biology. After careful consideration, we feel that it has merit but does not fully meet PLOS Computational Biology's publication criteria as it currently stands. Therefore, we invite you to submit a revised version of the manuscript that addresses the points raised during the review process.

Please submit your revised manuscript within 60 days Oct 14 2025 11:59PM. If you will need more time than this to complete your revisions, please reply to this message or contact the journal office at ploscompbiol@plos.org. Please include the following items when submitting your revised manuscript:

We look forward to receiving your revised manuscript.

Kind regards,

Varun Dutt, Ph.D

Academic Editor

PLOS Computational Biology

Mark Alber

Section Editor

PLOS Computational Biology

**Journal Requirements:**

At this stage, the following Authors/Authors require contributions: Eva Maxfield Brown, Stephan Druskat, Laurent Hébert-Dufresne, James Howison, Andrew Nesbitt, João Felipe Pimentel, Daniel Mietchen, and Boris A Veytsman. Please ensure that the full contributions of each author are acknowledged in the "Add/Edit/Remove Authors" section of our submission form.

2) The file inventory includes files for Figures 2a, and 2b. We would recommend either combining these into a single Figure 2.tiff file with separate internal panels, or renumbering them as individual figures, as we are not able to publish multiple components of a single figure as separate files.

**Reviewers' comments:**

Reviewer's Responses to Questions

Reviewer #4: In this article, the authors combine two metrics (popularity and centrality) to identify specific classes of biomedical open source software.

The authors rightly point out that software maintenance and funding is crucial to modern science. They add that their approach could be used to highlight the importance of some software packages and help their recognition and funding.

The authors use software from three ecosystems. PyPI, Cran, and Bioconductor. This covers a lot of the bioinformatics / -omics tools, but does not cover much of the imaging software used by other part of the community. While this is a limitation, I think it is ok to start with a subset of biomedical research software and let other communities apply the method on their own field. The authors provide the software to do so.

Disclaimer : while I am aware that the paper has been reviewed before, I was not in the first set of reviewers. The reviewers already covered important points.

While this is an interesting proof of concept, It is hard to estimate this work because, as any metric, this metric is largely arbitrary. This does not make it uninsightful, but I am not currently convinced. I have some questions & concerns before being able to fully assess the manuscript :

- Elephants in the room

My main surprise is to not see cornerstone libraries such as Blas & Lapack, or even higher level libraries such as numpy / scikit learn appear in this graph (while scikit-learn appears in the mentions dataset). In a way, there is nothing more central than those libraries ; as far as I know, most python scientific libraries will depend on numpy, that uses Blas & Lapack. I am surprised that they do not show up in the analysis.

Why is it so ? It is hard for me to trust the results knowing that.

- Cutoff

Where should the analysis stop ? In a way the code of the python/R interpreter, or of clang/gcc, is probably the most critical to this. Or even the windows OS and Linux Kernel ? Is there a cutoff somewhere ?

- Anaconda vs Pypi

Anaconda offers a very structured model for dependencies and distributing through conda-forge is much stricter on the dependencies than PyPI. Could looseness of the PyPi requirement system explain why obvious packages such as numpy/scipy/scikit lean do not appear ?

- The need for a metric :

Unfortunately science is very much metric-driven. While I believe the authors propose this in good faith, metrics come at a price as they can become decision tools (which is implied by the authors when they mention the importance of recognition for funding) ; then often the metric itself is optimized, instead of what it should measure(Goodhart's law). I nontheless think that this work is interesting out of curiosity.

- History vs current situation

As the authors mention, software is a living environment. Some libraries will disappear and others will appear. It seems that the author's analysis give a lot of weight on history, as evidenced by velvet, but also tophat. Tophat has not been updated for many years, nor is being actively forked recently. Is it still actually a cornerstone ?

More generally, how do you distinguish history from the current situation ?

Reviewer #5: Biomedical open source software: Crucial packages and hidden heroes is well-written and interesting. It addresses a very important topic of recognition and credit/incentives for open-source scientific software used—and frequently developed in—the academic setting. The traditional transactional nature of academic collaboration and authorship does not capture well software development efforts, let alone software maintenance efforts. Package and library dependencies in particular are easily taken for granted and their developers and maintainers are unsung heroes. In particular, researchers may rarely mention how they loaded data or performed basic numerical calculations. Thus, the goal of this paper to develop robust methods and metrics that can account for their role is very commendable and very important.

The traditional publishing side of academia has—for better or worse—the ubiquitous impact factors and h-indexes. Similar metrics for software contributions could help legitimize scientific software development efforts, particularly for early career researchers. I think these issues will only be further magnified both in the field of computational biology, but also in other scientific areas that may not been traditionally “computational”, with the growth of machine learning and challenges of big data.

To my knowledge, the approach here using Katz centrality is novel, not having been applied to the topic at hand. However, it is well motivated and explained in the text. While the dataset used is somewhat out of date at this point, this is not a major limitation, because the method should still provide insight into the dataset and then in the future be applied to the latest data. As a result, the work is of significant importance to the field and well suited for publication.

Major comments

I’m not very familiar with CRAN and Bioconductor, but I am quite familiar with the scientific Python ecosystem. As a result, I must say that I am quite puzzled and somewhat troubled by the lack of core packages like numpy, scipy, and matplotlib figuring anywhere in the results. I fully expected to see them in the “Pasteur” quadrant. Just taking numpy as an example, pymol, a “Pasteur” package, has numpy as a dependency, as does pace, a “Popular” package. Whether a result of the specifics of the underlying data set or some aspect of the methodology, I think it’s important that a package like numpy be accounted for somehow/somewhere in the text. Was it is simply considered a far outlier and as such discarded? Is it somehow a function of the age of the dataset (I have a hard time believing this)?

With that said, I was impressed that the package tifffile was identified as a “Nebraska” package! It indeed is the most powerful package for loading tiff files in Python (that I know of), but libraries used to load data, like images, are unlikely to be mentioned in a publication. Further, tifffile is literally developed and maintained by a single person, although I believe they are in California and not Nebraska.

Minor comments

- Page 3, The legend of Figure 1: it would be best to also have “Nebraska” and “Pasteur” quadrants briefly explained in the legend, to make the overall figure able to stand on it’s own. “Nebraska” in particular is quite a logical leap, even for those familiar with the XKCD cartoon.

- Page 3, “Nebraska” packages: it would be nice to connect back to the XKCD cartoon as mentioned earlier. In the earlier context on Page 2, the emphasis is on “thanklessly” and I had glossed over the “Nebraska” part of the quote. Thus, even as someone familiar with the cartoon, I was briefly confused by “Nebraska” on page 3. Alternately, consider on Page 2 to say that the term “Nebraska” will be used for such packages/libraries.

- Page 5, Results, a number of packages/libraries are mentioned here, but no references are provided, which is a bit ironic given the nature of the work. Some packages/libraries do get references later on, e.g. vctrs, reference 34, but I think all should be cited either in the text or in a table. This would also help fully disambiguate packages/libraries. For example, `tophat` on PyPI is a mobile app framework from 2012, so I have some doubt it that’s really the “Pasteur” package—perhaps it’s `top-hat` a “Recommendation system in TensorFlow”?

- Page 5, Results, “(a) unweighted graph, (b) weighted complete graph, and (c) the largest connected components of the weighted graph” — it would be helpful to clarify the differences between these three approaches. There is some interpretation of the differences in the Discussion, but here where these 3 models are first mentioned it would be helpful to clarify the differences in approach.

**Have the authors made all data and (if applicable) computational code underlying the findings in their manuscript fully available?**

Reviewer #4: None

Reviewer #5: Yes

PLOS authors have the option to publish the peer review history of their article (what does this mean?). If published, this will include your full peer review and any attached files.

Reviewer #4: No

Reviewer #5: No

**Figure resubmission:**
---

## [Decision Letter · Decision Letter 2]

12 Feb 2026

PCOMPBIOL-D-24-02238R2

Biomedical open source software: Crucial packages and hidden heroes

PLOS Computational Biology

Dear Dr. Howison,

Thank you for submitting your manuscript to PLOS Computational Biology. After careful consideration, we feel that it has merit but does not fully meet PLOS Computational Biology's publication criteria as it currently stands. Therefore, we invite you to submit a revised version of the manuscript that addresses the points raised during the review process.

We look forward to receiving your revised manuscript.

Kind regards,

Varun Dutt, Ph.D

Academic Editor

PLOS Computational Biology

Mark Alber

Section Editor

PLOS Computational Biology

**Journal Requirements:**

**Reviewers' comments:**

Reviewer's Responses to Questions

**Comments to the Authors:**

Reviewer #5: The Authors addressed my comments; however, after adding numpy, scipy, and napari to the analysis, they don't cite these packages when they appear in the text, e.g. page 12. Again, I find this very ironic given the nature of the work at hand. Please ensure that all of the packages discussed are cited properly!

Reviewer #6: As a new reviewer of this paper, I took time to reflect on the manuscript and the reviewer reports and responses. I agree with the other reviewers that the true core software libraries in scientific computing are things like BLAS, which are not covered by the data scope of the paper. At the same time I agree with the authors that their analysis nevertheless has value because it scopes a method and applies it to demonstrated networked software dependencies and risks within a narrower field of science. It contributes because it shows how immense value and interdependency emerges from the decentralized work of scientists. We should care about these libraries because, as the authors point out, incentives and of the maintainers in this part of the software stack is different from incentives in OSS coding in general.

Reviewer #7: I applaud the authors' efforts to improve this manuscript in previous rounds. As a new reviewer added just to this round, I don't think I have too many major comments, except that as a user of the CZI dataset, I have deep concerns of using this dataset for quantitative works. In our preliminary examination, we found pretty strong software name disambiguation problem, which was not captured by CZI's evaluation. I believe by focusing on just three software ecosystems, this problem can be alleviated, however, it cannot be totally solved. The one example that I hope the authors could check is Prism, which is a popular software name but is very poorly disambiguated in this dataset. So while the authors acknowledged this as a limitation, in my opinion, this is more than just a limitation that the authors should consider before designing this research.

A related comment is that, in my opinion, some more basic software, such as scipy and numpy, are way less frequently mentioned than they should be. This is an inevitable bias in scientific methods reporting: not every actually used software will be reported in the scientific papers. I hope the authors could reflect on this bias and talk about alternative ways to trace the impact of software (such as through citations) if they see fit.

**Have the authors made all data and (if applicable) computational code underlying the findings in their manuscript fully available?**

Reviewer #5: Yes

Reviewer #6: Yes

Reviewer #7: Yes

PLOS authors have the option to publish the peer review history of their article (what does this mean?). If published, this will include your full peer review and any attached files.

Reviewer #5: No

Reviewer #6: No

Reviewer #7: No

**Figure resubmission:**
---

## [Editor Report · Decision Letter 3]

20 Apr 2026

Dear Dr Howison,

We are pleased to inform you that your manuscript 'Biomedical open source software: Crucial packages and hidden heroes' has been provisionally accepted for publication in PLOS Computational Biology.

Best regards,

Varun Dutt, Ph.D

Academic Editor

PLOS Computational Biology

Mark Alber

Section Editor

PLOS Computational Biology

Based upon the revisions, the manuscript may be accepted in its current form.

---

## [Editor Report · Acceptance letter]

PCOMPBIOL-D-24-02238R3

Biomedical open source software: Crucial packages and hidden heroes

Dear Dr Howison,

I am pleased to inform you that your manuscript has been formally accepted for publication in PLOS Computational Biology. Your manuscript is now with our production department and you will be notified of the publication date in due course.

With kind regards,

Judit Kozma
